# Genome-Wide Association Study Identifies a Rice Panicle Blast Resistance Gene, *Pb2*, Encoding NLR Protein

**DOI:** 10.3390/ijms23105668

**Published:** 2022-05-18

**Authors:** Yao Yu, Lu Ma, Xinying Wang, Zhi Zhao, Wei Wang, Yunxin Fan, Kunquan Liu, Tingting Jiang, Ziwei Xiong, Qisheng Song, Changqing Li, Panting Wang, Wenjing Ma, Huanan Xu, Xinyu Wang, Zijing Zhao, Jianfei Wang, Hongsheng Zhang, Yongmei Bao

**Affiliations:** 1State Key Laboratory of Crop Genetics and Germplasm Enhancement, Jiangsu Collaborative Innovation Center for Modern Crop Production, Cyrus Tang Innovation Center for Crop Seed Industry, Jiangsu Province Engineering Research Center of Seed Industry Science and Technology, College of Agriculture, Nanjing Agricultural University, Nanjing 210095, China; 2018201030@njau.edu.cn (Y.Y.); 2019201068@njau.edu.cn (L.M.); 2019101063@njau.edu.cn (X.W.); 2020101064@stu.njau.edu.cn (Z.Z.); 2019201049@stu.njau.edu.cn (W.W.); 2020201035@stu.njau.edu.cn (Y.F.); 2019101062@njau.edu.cn (K.L.); 2020101065@stu.njau.edu.cn (T.J.); 2020101067@stu.njau.edu.cn (Z.X.); 2020801191@stu.njau.edu.cn (Q.S.); 2020101066@stu.njau.edu.cn (C.L.); xhn0409@163.com (H.X.); 11119403@njau.edu.cn (X.W.); 14219120@njau.edu.cn (Z.Z.); wangjf@njau.edu.cn (J.W.); hszhang@njau.edu.cn (H.Z.); 2School of Life Sciences and Biotechnology, Shanghai Jiao Tong University, Shanghai 200240, China; wangpanting1230@163.com; 3Department of Plant Science, School of Agriculture and Biology, Shanghai Jiao Tong University, Shanghai 200240, China; ma1083686270@163.com

**Keywords:** rice, panicle blast, genome-wide association study, haplotype analysis, rice diversity panel 1 (RDP1)

## Abstract

Rice blast is one of the main diseases in rice and can occur in different rice growth stages. Due to the complicated procedure of panicle blast identification and instability of panicle blast infection influenced by the environment, most cloned rice resistance genes are associated with leaf blast. In this study, a rice panicle blast resistance gene, *Pb2*, was identified by genome-wide association mapping based on the panicle blast resistance phenotypes of 230 Rice Diversity Panel 1 (RDP1) accessions with 700,000 single-nucleotide polymorphism (SNP) markers. A genome-wide association study identified 18 panicle blast resistance loci (PBRL) within two years, including 9 reported loci and 2 repeated loci (PBRL2 and PBRL13, PBRL10 and PBRL18). Among them, the repeated locus (PBRL10 and PBRL18) was located in chromosome 11. By haplotype and expression analysis, one of the Nucleotide-binding domain and Leucine-rich Repeat (NLR) *Pb2* genes was highly conserved in multiple resistant rice cultivars, and its expression was significantly upregulated after rice blast infection. *Pb2* encodes a typical NBS-LRR protein with NB-ARC domain and LRR domain. Compared with wild type plants, the transgenic rice of *Pb2* showed enhanced resistance to panicle and leaf blast with reduced lesion number. Subcellular localization of Pb2 showed that it is located on plasma membrane, and GUS tissue-staining observation found that *Pb2* is highly expressed in grains, leaf tips and stem nodes. The *Pb2* transgenic plants showed no difference in agronomic traits with wild type plants. It indicated that *Pb2* could be useful for breeding of rice blast resistance.

## 1. Introduction

Rice blast, one of the most widely distributed and destructive crop diseases, occurs not only in various parts of the world but also in different rice growth stages [1]. Panicle blast is a more severe form of rice blast and makes the grains imperfect and blackened, which will directly affect rice yield and quality. However, it is difficult to study, due to difficulty in the phenotypic identification affected by many environmental factors. To date, at least 38 rice blast resistance genes were cloned [2,3,4,5]. However, only four genes among them were reported to have panicle blast resistance. *Pb1* is a disease-resistant gene cloned from the indica rice cultivar Modan, which confers unique resistance to panicle blast, with higher expression level at the heading stage than the seedling stage [6]. *Pi64* is the panicle blast and leaf blast resistance gene cloned from japonica rice Yangmaogu varieties and is expressed constitutively in all rice growth stages and different tissues [7]. *Pid3-A4*, a homolog of *Pid3*, was cloned from the common wild rice A4 (*Oryza rufipogon*), and it is constitutively expressed and provides resistance to both leaf blast and panicle blast [8,9]. *Pigm* is a panicle and leaf blast resistance gene cloned from Gumei 4, and the expression level is high in anthers. This NLR protein pair Pigm-R and Pigm-S can balance the disease resistance and yield mechanisms through epigenetic regulation [10].

Plant resistance (R) genes encoding NLR proteins with nucleotide binding sites (NBS) and leucine-rich repeat (LRR) domains play important roles in plant defense systems [11,12]. The NBS domain belongs to the STAND (signal transduction ATPases with numerous domain) superfamily and might participate in disease resistance signaling as molecular switch [13]. The LRR domain of plant NBS-LRR proteins is involved in specific recognition of pathogen effector molecules [14]. Pi-ta can interact with the expression product of the avirulence gene of *AVR-Pita* of *M. oryzae* to induce disease resistance [15]; Pit mediates ROS production and HR responses by directly interacting with the GTPase OsRac1 at the plasma membrane [16]. Occasionally, NLRs also regulate plant defense responses in gene pairs. These two indispensable genes increase resistance of the plant to the disease, including *Pi-5* [17], *Pi-km* [18], *Pi-k* [19], *Pi-a* [20] and *Pi-1* [21]. NLRs gene pairs also achieve a balance between resistance and yield through reciprocal regulation of each other. It has been proven that PigmS competitively attenuates PigmR homopolymerization to suppress resistance, and *PigmS* improves seed production to counteract the yield cost caused by *PigmR* [10].

The genome-wide association study (GWAS) is based on linkage disequilibrium to detect the genetic variation (marker) polymorphisms of multiple individuals in the whole genome to obtain the genotype associated with the observable traits. Compared with the traditional QTL mapping method, it has the advantages of high resolution, a wide range of research materials, rich captured variations, and high efficiency to locate multiple traits simultaneously. After more than ten years of development, GWAS technology has been widely used in important agronomic traits of rice, such as ideal plant architecture, grain shape, grain weight, and flowering period [22,23,24]. Lu et al. (2015) detected 298 ideal plant architecture loci by GWAS on the compact plant type, short stature, few unproductive tillers, thick and sturdy stems and erect leaves of 523 rice materials [25]. Duan et al. (2017) detected three grain width-related loci by GWAS on the grain width of 102 indica rice materials [24]. In recent years, some progress has been achieved in the application of GWAS in rice disease resistance [26,27]. Zhang et al. (2017) identified 12 bacterial blight resistance-related loci by performing GWAS of 172 rice germplasms [26]. Zhang et al. (2019) identified 27 sheath blight resistance-related loci by performing GWAS of 563 rice germplasms [27].

A large number of rice blast resistance loci have been identified by GWAS, which accelerates the cloning of rice blast resistance genes. Five rice blast resistance loci including the cloned gene *Pita* were first identified by GWAS, which verified the feasibility of identifying rice blast resistance loci by GWAS [28]. To date, in total more than 230 rice blast resistance-related loci distributed in all 12 chromosomes were identified by GWAS [29,30,31,32,33,34,35]. Among them, 16 resistance loci were identified through naturally occurring resistance phenotypes in the blast nurseries, which can resist the invasion of various strains [31]. The candidate gene types, including protein kinases, transcription factors, ubiquitin-related, phosphorylation-related, DNA/ATP binding, oxidase/oxidoreductase and heat shock proteins, were obtained, which provided the possibility for cloning new types of resistance genes other than NLRs [29,31,32,34]. In addition, genome-wide associations between rice blast resistance and yield-related components revealed a complex relationship between disease resistance and yield-related components [30]. GWAS of rice blast resistance under different nitrogen concentrations revealed the cause of nitrogen-induced susceptibility [36]. Although a large number of resistance-related loci have been located by the GWAS, few new genes have been functionally characterized well. The pair of blast resistance genes *LABR_64-1* and *LABR_64-2* cloned from the natural population by GWAS are *Pi5* alleles [32]. The *bsr-d1* is a gene cloned by GWAS with a population of RILs, and *bsr-d1* is a C_2_H_2_-type transcription factor that inhibits *M. oryzae*-induced expression of *Bsr-d1* RNA and degradation of hydrogen peroxide to achieve resistance to *M. oryzae* [3].

The main goal of this study was to find the panicle blast resistance gene. In previous studies, RDP1 has been widely used in rice blast resistance research, and more than 180 leaf blast resistance-related loci have been identified, of which 16 were identified by natural occurrence in the field [28,31,32,33,34]. However, there is no report on the identification of panicle blast resistance using RDP1. This study provides important clues for the identification and cloning of rice blast panicle genes and provides an important basis for panicle resistance breeding by identifying RDP1 in the field and performing PBRL scanning with GWAS.

## 2. Results

### 2.1. Panicle Blast Resistance Evaluation of RDP1

According to the average percentage of diseased grains, the incidence of panicle blast was evaluated into 10 grades (Figure 1A). The identification results for two consecutive years show that 62.9–63.7% of the materials had a percentage of diseased grains below 10%, and 81.7–84.2% of them were less than 20%. The average percentage of diseased grains were 12.46% and 11.03%, respectively. The population material was biased towards disease-resistant rather than susceptible (Figure 1C,D). In the identification results of panicle blast in 2018, the percentage of diseased grains of ADMIX, AUS, IND, TEJ and TRJ above 10% were 30.30%, 29.27%, 25.00%, 47.17% and 30.77, respectively (Figure 1E). In the identification results of panicle blast in 2018, the percentage of diseased grains above 10% were 40.62%, 19.44%, 18.18%, 46.94% and 36.73%, respectively (Figure 1F). The analysis results showed that the two-year blast resistance was positively correlated (r = 0.41, *p* < 0.0001).

### 2.2. Identification of Panicle Blast Resistance Loci in Rice Genome

In total, 18 panicle blast resistance loci (PBRL) were identified in two years, which were distributed on chromosomes 1, 3, 4, 6, 9 and 11, including 9 reported loci and two repeated loci (Figure 2A,C and Table 1). A total of 10 PBRLs, including 4 PBRLs on chromosome 11, were identified in 2018 (Figure 2A). In addition to the reported R genes or QTLs, nine new loci on chromosome 1 (PBRL-1, PBRL-11), chromosome 3 (PBRL-2, PBRL-13), chromosome 6 (PBRL-15, PBRL-16), chromosome 9 (PBRL-17) and the short arm of chromosome 11 (PBRL-7, PBRL-8) associated with panicle blast resistance were detected. Among these new loci, one repeated site (PBRL-10/PBRL-18) with the highest peak value (−log_10_*P* = 9.08 and −log_10_*P* = 6.46) was identified at the end of the long arm of chromosome 11.

In order to further analyze the candidate genes in the repeated locus (PBRL10 and PBRL18), a 1.47 Mb intersection containing PBRL10 and PBRL18 was selected for LD block heatmap analysis. Significant SNP (−log_10_*P* ≥ 4) density distributions paired with LD (r^2^ ≥ 0.6) triangular blocks results, and the candidate region of the 142.2 Kb interval between SNP_27581700-SNP_27723865, were finally determined (Figure 2E). The ORFs in the 142.2 Kb interval were predicted by the gene annotation network RGAP (http://rice.uga.edu/, accessed on 14 November 2019), and a total of 22 candidate genes were predicted, including 1 gene encoding LRR proteins, 2 genes encoding NB-ARC proteins, 1 gene encoding CC-NBS-LRR proteins, 1 gene encoding WARK125, 1 gene encoding MYB transcriptional regulator, 1 gene encoding galactosyltransferase, 2 genes encoding pectinesterases, 6 genes encoding transposons, 1 gene encoding retrotransposon and 6 genes encoding expressed proteins (Appendix A).

### 2.3. Screening and Determination of Candidate Genes

According to the method of Yano et al. (2016) [37], all SNPs in the 142.2 Kb candidate region were classified into five groups (Appendix A). The selection of candidate genes is performed mainly from Group I, which includes polymorphisms (−Log_10_*P* ≥ 4) that were significantly associated with GWAS trait variation and predicted to induce amino acid exchange or altered splicing junctions of introns (GT or AG at the start or end of intron, respectively). In this study, in order to avoid missing important candidate genes, the candidate genes in the second group were also screened, which contains polymorphisms that were significantly associated with trait variation in Group I and were located in the 5′ UTR flanking sequence of the gene (before the first ATG and ≤2 Kb, such as the promoter region). Among the 229 SNPs in the candidate region, only two SNPs (SNP_27597071 and SNP_27597338) belong to Group I and fall on the same gene LOC_Os11g45600, which encodes a resistance protein (Appendix A). Four SNPs belong to Group II and land on the promoters of four genes and the 5′ UTR of one gene (Appendix A). Among them, three genes including LOC_Os11g45600 encoding LRR protein, LOC_Os11g45720 encoding pectinesterase and LOC_Os11g45740 encoding MYB family transcription factor were considered as the most likely candidate genes. Haplotype analysis showed that only the candidate gene LOC_Os11g45600 in Group I had haplotype differences in the panicle blast phenotypes for two years (Figure 3A,B; Appendix A).

To prevent the omission of important genes, we also analyzed all SNPs in the 142.2 Kb interval, and 11 genes were identified with amino acid changes due to differences in SNPs (Appendix A). The haplotype analysis on these 11 genes showed that only three genes had significant differences in the two-year panicle blast identification (Figure 3A–F and Appendix A). The analysis of the expression patterns of these three genes and the expressions of LOC_Os11g45600 and LOC_Os11g45620 both increased significantly at 48 hpi with 28.92 and 27.69 folds (Figure 3G), while the expression of LOC_Os11g45710 increased at 8 h and 72 h with 1.9 and 2.7 folds changes, respectively (Figure 3I). LOC_Os11g45710 encoded glycosyl hydrolase protein, which was rarely reported in fungal disease resistance. It was intriguing that LOC_Os11g45620 encoded for LRR protein, LOC_Os11g45600 encoded for NB-ARC protein and their expression patterns were almost the same (Figure 3G,H). These two genes were suspected to be the same gene. According to the haplotypes analysis, there were three haplotypes (Hap.1, Hap.2, and Hap.3) of LOC_Os11g45600 divided by two significant SNPs (Figure 3J). Hap.1 was associated with resistant, and Hap.3 was associated with susceptible, whereas Hap.2 was the intermediate type (Figure 3K).

### 2.4. Cloning of Panicle Blast Resistance Gene Pb2

The Taihu Lake region has a long history of rice cultivation and thus contains various Japonica rice landraces, including the resistant cultivar Jiangnanwan and susceptible cultivar Suyunuo. SNP analysis showed that Jiangnanwan corresponded to the resistant haplotypes Hap.E or Halp.1in LOC_Os11g45600, whereas Suyunuo was prematurely terminated due to a single-base A to T (+1291 bp) transition. Based on the sequence analysis results, the gene LOC_Os11g45600 was part of the last exon of LOC_Os11g45620, and the gene LOC_Os11g45610 encoding retrotransposon was deleted in both Jiangnanwan and Suyunuo (Figure 4A). To test whether this was a common phenomenon in other cultivars, the presence of LOC_Os11g45610 in Nipponbare and 232 were verified as natural populations by PCR amplification with two pairs of primers. The results showed that LOC_Os11g45600 and LOC_Os11g45620 fused together to constitute a complete CC-NBS-LRR type gene named *Pb2* (Figure 4A,B), and 99% of the accessions did not have the retrotransposon LOC_Os11g45610 (Figure 4C). Amino acid sequence alignments of Pb2 showed that there were multiple amino acid differences between Pb2-J and Pb2-S (Figure 4B). Although there was no difference between *Pb2-S* and *Pb2-R* in the significant SNP_27597071 and SNP_27597338, the conversion of single base A to T (+1291 bp) led to the termination of *P2-S*, lacking an LRR domain (Figure 4B).

### 2.5. Specific Expression Analysis of Pb2

Intracellular localization of Pb2 was detected by observing the root tip cells of the *P_Pb2_::RFP-Pb2* fusion transgenic plants, and it was found that Pb2 was mainly localized on the plasma membrane (Figure 5A). The expression of *Pb2* was induced after being inoculated by blast fungi at the booting stage, but *Pb2-J* in resistance cultivar Jiangnanwan was induced more obviously than in the susceptible cultivar Suyunuo (Figure 5B). In addition, Pb2 was specifically expressed in coleoptiles, stem nodes, leaf tips and grains by GUS staining of *P_Pb2_::Pb2-GUS* transgenic plants (Figure 5C).

### 2.6. Pb2 Associated with Panicle and Leaf Blast Resistance

In order to verify the function of *Pb2* in rice blast resistance, complementation, overexpression and knockout, transgenic lines of *Pb2-J*, which was cloned from resistant cultivar Jiangnanwan, were constructed. When *Pb2-J* was complementary to the susceptible material Suyunuo, *Pb2-J* could effectively reduce the percentage of diseased grains and improve panicle blast resistance. However, the panicle blast resistance of *Pb2-J* was relatively moderate resistance (Figure 6E–G and Appendix A). The overexpression in transgenic plants of *Pb2-J* in Suyunuo also showed enhanced panicle blast resistance (Figure 6E–G and Appendix A). Compared with the resistant cultivar Jiangnanwan, the knocked out transgenic plants of *Pb2-J* in Jiangnanwan showed more susceptibility in 2020 Nanjing, while only part of the line had a difference detected in other trials (Figure 6A–D and Appendix A). The complementary and overexpression of *Pb2-J* transgenic plants also showed obviously enhanced resistance to blast in the seedling stage (Figure 6H–J and Appendix A). There was no difference between complementary, overexpression and knockout in the transgenic lines and their wild-type (WT) plants in the agronomic traits, including plant height, number of effective tillers, panicle length, filled grains per panicle, seed setting rate, 100-grain weight, grain length and grain width (Figure 7A–J).

## 3. Discussion

Compared with traditional map-based cloning, GWAS offers higher mapping efficiency with a greater number of loci and many polymorphic SNPs. GWAS is becoming widely applied for identifying target genes, and the suitable population structure, effective SNP density and the accuracy of phenotypic identification are the basis of GWAS [28,29,38,39]. In our research, RDP1 was selected as the population for GWAS analysis, which had a high density of 700 K SNPs. Previous studies have shown that RDP1 had a large number of rice seedling plants blast resistance-related loci, and it confers resistance to blast isolated from China, South Korea, Colombia, Philippines, India and America. Moreover, RDP1 had high-level adult plants blast resistance in the field disease nursery in rice producing areas of China (Fujian Province, Hunan Province and Heilongjiang Province) [28,31,32,34]. For the field panicle blast phenotype identification, an injection inoculation method was utilized and the diseased grains percentage was observed [40]. The identification results indicated that AUS and IND had better resistance to panicle blast, and TEJ had the worst resistance to panicle blast. Although the resistance results of panicle blast were identified in the field and were easily affected by environmental factors, in this study, the results of correlation analysis showed that the identification results of the two years were positively correlated.

Panicle blast on rice is irreversible and consequently reduces the quality and grain yield of rice [41]. Research on panicle blast is difficult due to factors such as heavy workload, long research period and uncontrollable environmental conditions. Many rice blast resistance loci have been identified by GWAS [31,32,33,34,35]. However, there is no report for the identification of panicle blast resistance loci by GWAS. Leaf blast and panicle blast occur in different rice growth stages, and their resistance mechanisms are not completely consistent [6,42]. In this study, eighteen panicle blast resistance-related loci were identified by GWAS on two-year repeated data. Seven resistance genes at these four PBRLs have been reported, with the details as PBRL-4 (*Pid3*, *Pi25*), PBRL-10/PBRL-18 (*Pik*, *Pikm*, *Pikp*, *Pi1*) and PBRL-14 (*Pi21*) [19,43,44]. A total of 6 PBRLs, PBRL-3 (LABR-45), PBRL-6 (LABR-68), PBRL-9 (LABR-82, LABR-83), PBRL-10/PBRL-18 (LABR-84, LABR-85) and PBRL-12 (LABR-32), were co-localized with previously identified QTLs in GWAS analysis of RDP1 [32]. GWAS is an efficient strategy to save time and detect more micro-effective sites compared with traditional map-based cloning. However, GWAS also has limitations in quickly narrowing the candidate interval and accurately obtaining candidate genes [37]. To date, few rice blast resistance genes have been cloned by GWAS. Liu et al. (2019) [45] identified a new non-strain-specific partial R gene *PiPR1* on chromosome 4 by performing leaf blast resistance identification and GWAS on the C-RDP-II germplasm population. *PiPR1* is an NLR gene that is highly conserved in multiple partially resistant rice cultivars, and its expression is significantly upregulated at the early stages of rice blast infection [45]. In this study, NLR gene *Pb2* was successfully cloned by bioinformatics and expression analysis of the repeat sites PBRL-10 and PBRL-18. Genetic complementation and overexpression analyses verified that it confers panicle blast and leaf blast resistance. A large gene cluster, which was critical to complete blast resistance, has been located at the terminal long arm of chromosome 11 [19,21,46]. Multiple NLR genes at the PBRL-10 locus were identified, and *Pb2* may be a member of the gene cluster.

Rice blast R genes have been extensively used in rice resistance breeding [47]. R genes such as *Pigm*, *Pi2*, and *Pi9* have broad-spectrum and durable rice blast resistance; however, with continuous planting of cultivars with these R genes, the resistance may be overcome by *Magnaporthe oryzae* fungus after evolution and variation [48,49,50,51]. In contrast, partial R genes have been associated with durable resistance [52]. Therefore, identification and cloning of partial R genes will be of great significance in rice breeding. So far, four partial blast resistance genes, *pi21*, *Pi35*, *Pb1*, and *PiPR1,* have been cloned in rice [6,44,45,53]. In this study, the partial resistance gene *Pb2* and seven new panicle blast resistance-related loci were identified by GWAS, providing potential application in rice resistance breeding. Several studies have shown that R genes were differentially expressed between resistant and susceptible genotypes, which were induced by *M. oryzae* and mediated resistance responses [54,55,56]. In this study, *Pb2* was induced by *M. oryzae* and exhibited a 28.9-fold increase in expression in rice panicles after 48 h of inoculation. This result implied that *Pb2* would mediate panicle blast resistance. Higher expression levels were detected in the leaf tips and grains of transgenic plants by GUS staining. This may explain the reason for *Pb2* conferring both leaf and panicle blast resistance. In addition, panicle R gene *Pb2* was also found to mediate penetration resistance in the early stages of *M. oryzae* invasion, as it could only reduce the number of lesions but not the length of lesions. Pb2 was localized to the plasma membrane and has a short LRR domain (two LRRs). The LRR domain of plant NBS-LRR proteins was involved in specific recognition of pathogen effector molecules [14]. It has been reported that defense activation of R proteins such as MLA10, AtRPS4 and AtRPS6 requires nuclear localization or translocation to the nucleus [57,58,59,60].

During the resistance process, plants will trigger a series of defense responses, which will affect the energy distribution for plant growth [61,62]. Therefore, improving crop resistance is often accompanied with a loss in yield and grain quality [63,64,65]. Plant defense mechanisms have evolved during the coevolutionary arms race between plant and pathogen, which finely regulate disease resistance and balance growth and development. PigmR confers broad-spectrum resistance, whereas PigmS competitively attenuates PigmR homodimerization to suppress resistance. *PigmS* increases grain production to offset the yield cost caused by *PigmR* [10]. Phosphorylated IPA1 binds to the promoter of the pathogen defense gene *WRKY45* to enhance disease resistance, but IPA1 rapidly reverts to a non-phosphorylated state after infection, maintaining growth required for high yields [5]. The rice calcium-binding protein ROD1 maintains normal rice growth by promoting ROS scavenging, but is degraded by two E3 ligases RIP1 and APIP6 induced by pathogen infection, maintaining the ROS content required for immune activation [66]. *Pb2* increased resistance without affecting agronomic traits. This balance may be due to the low expression level of *Pb2* in normal growth, which increased rapidly after infected with *M. oryzae* but quickly returned to normal levels after 72 h.

## 4. Materials and Methods

### 4.1. Plant Materials and Growth Conditions

The Rice Diversity Panel 1 (RDP1) containing 413 *Oryza sativa* accessions from 82 countries was provided by Cornell University, USA, and it has 700 K high-quality SNPs, representing a wide range of genetic variations in *Oryza sativa* [28]. According to the growth period, 230 accessions were selected from RDP1: including 44 Indica (IND), 55 Temperate Japonica (TEJ), 51 Tropical Japonica (TRJ), 42 AUS, 7 AROMATIC, and 33 other accessions classified as admixtures (ADMIX) between Temperate and Tropical Japonica groups. Two landraces, Jiangnanwan and Suyunuo from the Taihu Lake region, preserved by the State Key Laboratory of Nanjing Agricultural University were added to RDP1 (Appendix A).

The plants grow in fields at the Nanjing Experimental Base in Jiangsu province and the Sanya Experimental Base in Hainan Province. Panicle blast inoculation was carried out by injection method when the plants grow in the fields at the booting stage of plants. The leaf blast resistance inoculation was carried out in the greenhouse in Nanjing Agricultural University. The seeds were sown in plastic seedling trays (60 cm × 30 cm × 5 cm) filled with organic soil, 6–8 seedlings per hole [67]. Seedlings were grown in a greenhouse at 22–30 °C, 14 h light/27 °C, 10 h dark/22 °C.

### 4.2. Blast Inoculation and Disease Evaluation

The *Magnaporthe oryzae Hoku1* provided by the Institute of Plant Protection of Chinese Academy of Agricultural Sciences was used for inoculation. For the panicle blast resistance identification, a 2 mL spore suspension with a concentration of 1 × 10^5^/mL was used for injection at the booting stage of rice. A total of 9 panicles of each line were injected with 3 strains. The percentage of diseased grains was calculated after three weeks. Grains infected with spore-forming areas were defined as diseased grains [40].

For the leaf blast resistance identification, a spore suspension with a concentration of 1 × 10^5^/mL was used, which was combined with 0.02% Tween 20 (Solarbio, Beijing, China, Cat#T8220) [68,69]. According to the method of Zhu et al. (2012) [68], each seedling tray is inoculated with 60 mL of spore suspension by spraying the seedlings with a spray gun connected to an air pump. After inoculation, seedling trays were placed in a dark environment at 95–100% relative humidity and 27 °C for 24 h and then transferred to a greenhouse with the temperature maintained at 27 °C. The length and number of lesions were recorded after six days. Six plants for each line and three replicates were measured, and the average was calculated for the resistance phenotype.

### 4.3. Genome-Wide Association Study

Using Tassel 5.0 (http://www.maizegenetics.net/tassel/, accessed on 16 April 2022) [70], the 700 K high-density SNPs of RDP1 were filtered, and filtered 556,809 SNPs (MAF > 0.05, missing rate < 25%) were used for GWAS. The file format was converted with PLINK software (http://pngu.mgh.harvard.edu/purcell/plink/, accessed on 16 April 2022) [71], and the hapmap files were converted into tped and tfam formats for association analysis. The EMMAX analysis online script was obtained from the website (http://genetics.cs.ucla.edu/emmax/install.html, accessed on 16 December 2019) [72]. Under the Linux environment, a file containing the confidence value (*p*-value) of each SNP on the whole genome was created. Manhattan plots and Quantile-quantile (Q-Q) plots were calculated by qqman software package in the R language for plotting (R Development Core Team 2011). The threshold *p*-value < 10^−4^ was chosen to obtain significantly related SNPs, and then more than three consecutive SNPs (the distance between SNPs were less than 200 kb) were named as related loci [73].

### 4.4. RNA Isolation and qPCR Analysis

Panicle samples were collected at 0 hpi, 2 hpi, 4 hpi, 8 hpi, 12 hpi, 24 hpi, 48 hpi and 72 hpi after being inoculated by blast fungi and quickly frozen in liquid nitrogen. Three biological replicates were conducted. TaKaRa MiniBEST plant RNA extraction kit (Kusatsu, Japan, TaKaRa Code: 9769) was used for RNA extraction. 200 ng RNA was reverse transcripted to cDNA by HiScript^®^ II RT SuperMix for qPCR (Nanjing, China, Vazyme Code: R223-01) and was uniformly diluted 10 times as a template for qRT-PCR fluorescence quantitative analysis. The primers in Appendix A were amplified for the genes’ expression analysis, and *OsActin1* (LOC_Os03g50885) was invoked as the internal control (Actin-QF/R). qRT-PCR was performed using AceQ^®^ qPCR SYBR Green Master Mix (Nanjing, China, Vazyme Code: Q111-02) and read with the LC480 II (Roche fluorescence quantitative LightCycler480, San Francisco, CA, USA) instrument. Reactions were set up as follows: 5 min at 95 °C, followed by 40 cycles of 95 °C for 10 s, 60 °C for 30 s, and 72 °C for 40 s. The relative expression level of each gene was calculated using the 2^−^^△△CT^ method. Three biological replicates were performed for each qRT-PCR.

### 4.5. Plasmid Construction

For the complementation of *Pb2-J*, excision of the 2 × 35S promoter of the binary vector pCAMBIA1300s by Hind *III* and Xba *I*, the genomic fragment containing the 3276 bp promoter and the 3792 bp entire gene was cloned into the binary vector pCAMBIA1300s without the 2 × 35S promoter. pCAMBIA1300s was digested with Kpn *I* and BamH *I* double enzymes, the entire genome sequence (3792 bp) of *Pb2-J* was cloned into the vector pCAMBIA1300s containing the 2 × 35S promoter to generate overexpression lines. For the promoter activity assay of *Pb2*, the binary vector pCAMBIA1301 was digested with BamH *I* and Xba *I* enzymes, and the fragment containing 3276 bp promoter sequence was introduced into the binary vector pCAMBIA1301, which can promote the GUS gene’s expression. The promoter and the CDS of *Pb2* were cloned and inserted into the binary vector pCAMBIA1300s-mCherry to generate *P_Pb2_::RFP-Pb2* for subcellular localization. To generate knockout expression of *Pb2* transgenic plants, the vector pBUE411-2gR (provided by Chen Qijun, College of Biology, China Agricultural University) was selected for construction. The targets designed for the NB-ARC domain were as follows: target 1: 5′-GGCAGCTCTAGGCTCTAAT-3′; target 2: 5′-CCAATATCTGATCTGGAAG-3′. The genetic transformations were conducted using rice embryogenic calli through *Agrobacterium tumefaciens*-mediated transformation. All endonucleases were provided by America New England Biolabs, high-fidelity enzyme Phanta® Max Super-Fidelity DNA Polymerase (Vazyme Code: P505-d1) and homologous recombinase ClonExpress® II (Vazyme Code: C112-02) were provided by China Vazyme.

### 4.6. Subcellular Localization

The seeds of *P_Pb2_::RFP-Pb2* transgenic plants (T0) and Suyunuo (WT) were soaked in water for two days and germinated at 27 °C for one day, and then healthy seeds were selected and planted in 0.3% agar gel (Solarbio, Cat#A8190) containing 50 μg/mL hygromycin B (Solarbio, Cat#H8080). After culturing for 4 days in an artificial climate box (dark at 27 °C), plants with normal root growth were selected and the roots were rinsed with ddH_2_O. The 2–3 mm root tip was chosen to make slices, and the fluorescence value was observed under a laser confocal microscope (Leica SP8, Heidelberg, Germany). The excitation wavelength was 580 nm and the emission wavelength was 610 nm. In order to avoid false fluorescence signals due to autofluorescence, first, 5–7 samples with WT as the control were observed, and then the fusion RFP material was observed after confirming that there was no autofluorescence.

### 4.7. Pb2Pro::GUS Staining Assay

Different tissues, such as root, leaf, leaf sheath at seedling stage and root, stem, stem node, leaf, leaf sheath, young panicle grain at heading stage from *Pb2Pro::GUS* transgenic plants and WT, were collected and incubated in a 50 mL centrifuge tube with GUS staining solution (50 mmol/L PBS, 2 mM K_3_Fe(CN)_6_, 2 mM K_4_[Fe(CN)_6_] 3H_2_O, 10 mmol/L Na_2_EDTA, 0.1% Triton-X100, 20% methanol, 1 mg/mL X-Gluc) immersion. The samples were evacuated to −0.1 MPa for 20 min and were placed in the dark at 37 °C for 12 h, and then were destained 3–4 times with 95% ethanol until the control turned white. Finally, the GUS staining photos were taken with a stereoscope.

### 4.8. Data Analysis

A 700-K dataset and efficient mixed-model association were used for GWAS analysis and significant SNPs were selected −log_10_*P* > 4 [74]. Haplotype analysis was performed on all SNPs that fell on the gene (including non-significant SNPs). GraphPad Prism 8.0 software (Provided by GraphPad Software, San Diego, CA, USA) was used for data analysis; a *t*-test at the 0.05 level and 0.01 level and least significant difference (LSD) method were used for difference comparison analysis; SPSS 19.0 software (Provided by IBM, New York, NY, USA) was used for correlation analysis.

## Figures and Tables

**Figure 1 ijms-23-05668-f001:**
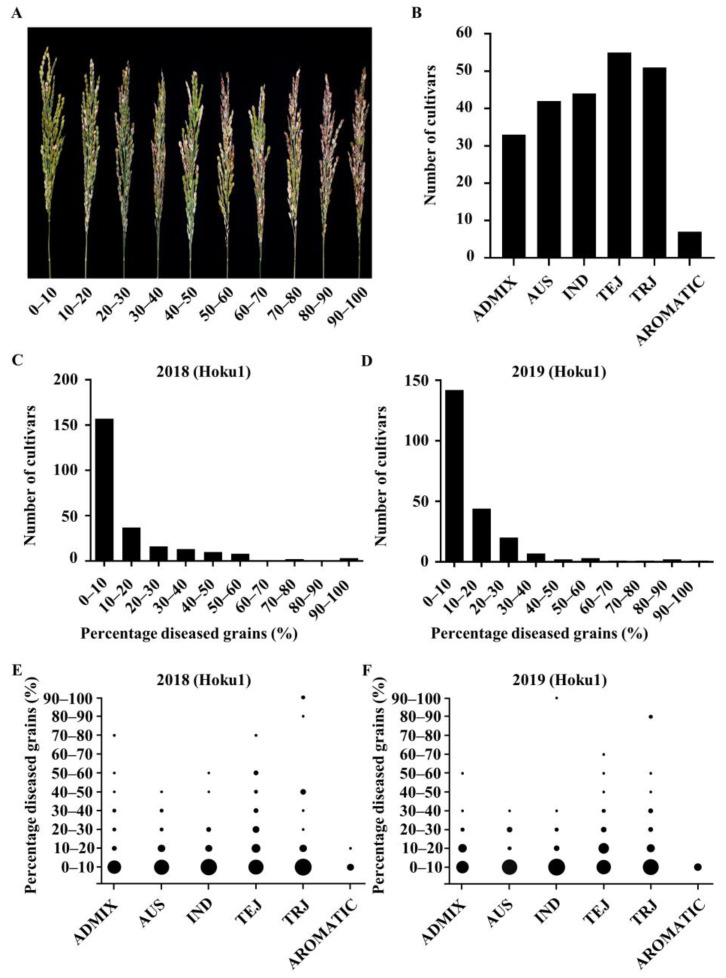
Grouping and identification of panicle blast resistance. (**A**) Identification and grading of panicle blast resistance. (**B**) Distribution of 232 rice germplasms in subgroups. IND: indica; TEJ: tropical japonica; TRJ: temperate japonica; AUS: Autumn Rice, which can be classified as indica; ADMIX: mixed variety; AROMATIC: aromatic variety, which can be classified as japonica. (**C**,**D**) Frequency distribution of panicle blast resistance scores in fields at the Nanjing Experimental Base in Jiangsu province in 2018 and 2019, respectively. (**E**,**F**) Distribution of blast resistance scores of RDP1 in the six sub-populations; the area of black circles represents the accession numbers.

**Figure 2 ijms-23-05668-f002:**
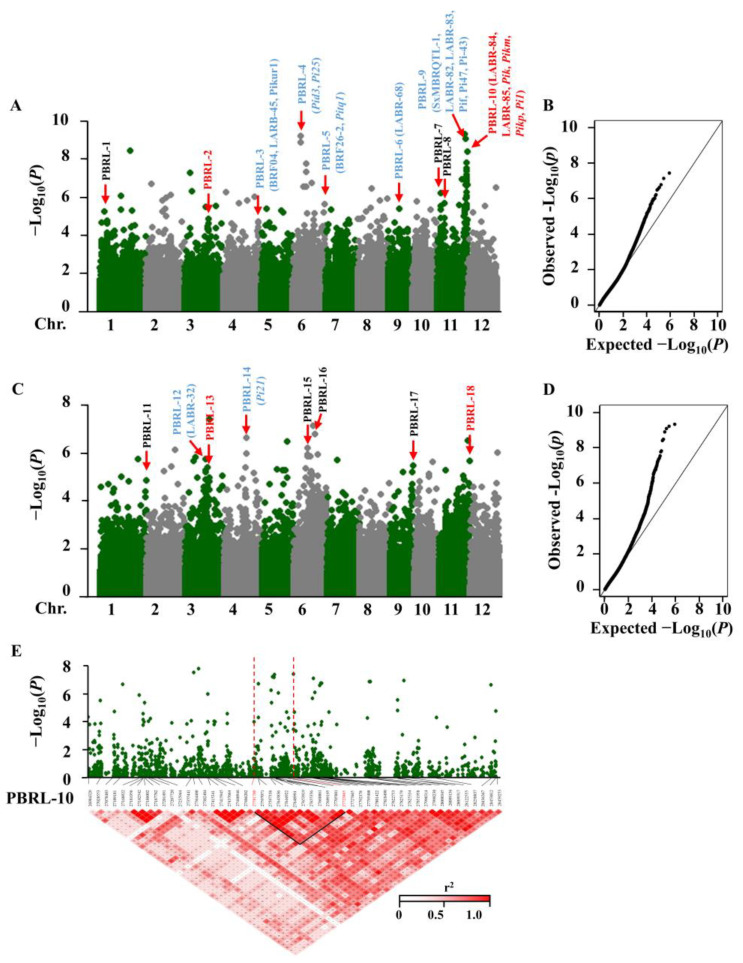
Genome-wide association study of panicle blast resistance. (**A**,**C**) Manhattan plots of markers associated with rice blast resistance to panicle blast; the X-axis is the location of each single nucleotide polymorphism (SNP) on the 12 chromosomes, and the Y-axis is the *p* value obtained from the GWAS model −Log_10_*P*. SNPs with strong associations with PBRLs had higher Y-coordinate values. The reported genes or loci are marked in blue font; the PBRLs identified by two-year data repeats are marked in red font; newly identified non-repeated PBRLs are marked in black font. (**B**,**D**) Quantile-quantile (Q-Q) plots showing the fitness of selected models for different traits across populations or subgroups. X-axis: expected −Log_10_*P*; Y-axis: observed −Log_10_*P*. (**E**) Local Manhattan plot (top) (27.0–28.5 Mb) and LD heatmap (bottom) (27.0–28.5 Mb) around the peak of PBRL-10.

**Figure 3 ijms-23-05668-f003:**
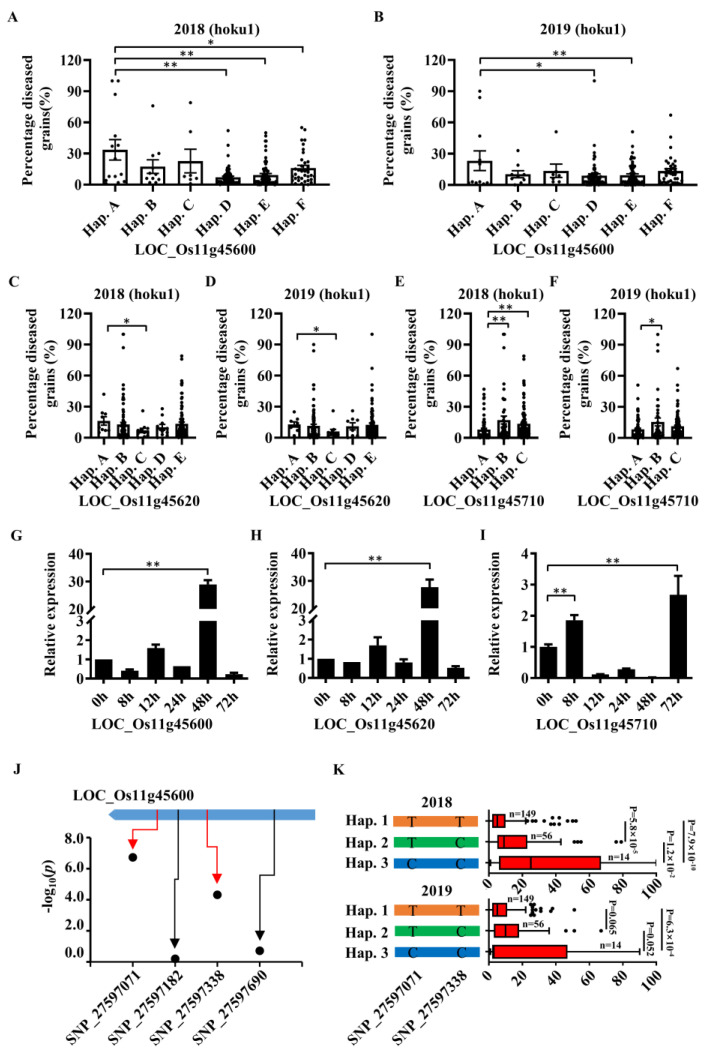
Genes with significant haplotype differences are induced by *Magnaporthe oryzae*. (**A**–**F**) Percentage diseased grains (%) caused by different haplotypes. Presented as a scatter plot with bars, the edge of the box represents the mean, and each point represents the percentage diseased grains for a germplasm. (**G**–**I**) The expression patterns of candidate genes (LOC_Os11g45600, LOC_Os11g45620 and LOC_Os11g45710) in the disease-resistant material Jiangnanwan inoculated with *M. oryzae Hoku1* were studied by qPCR. The amplification of *OsActin1* (LOC_Os03g50885) was used as an internal reference. Error bars are the standard deviation of three technical replicates. (**J**) Location of all SNPs on LOC_Os11g45600 and their correlation with trait variation. Significant (−Log_10_*P* ≥ 4) SNP positions are marked with red arrow lines. (**K**) Haplotype analysis of two significant SNPs on LOC_Os11g45600; the “n” represents the number of rice varieties in the population. Results are represented as the mean ± SEM; (* *p* < 0.05, ** *p* < 0.01).

**Figure 4 ijms-23-05668-f004:**
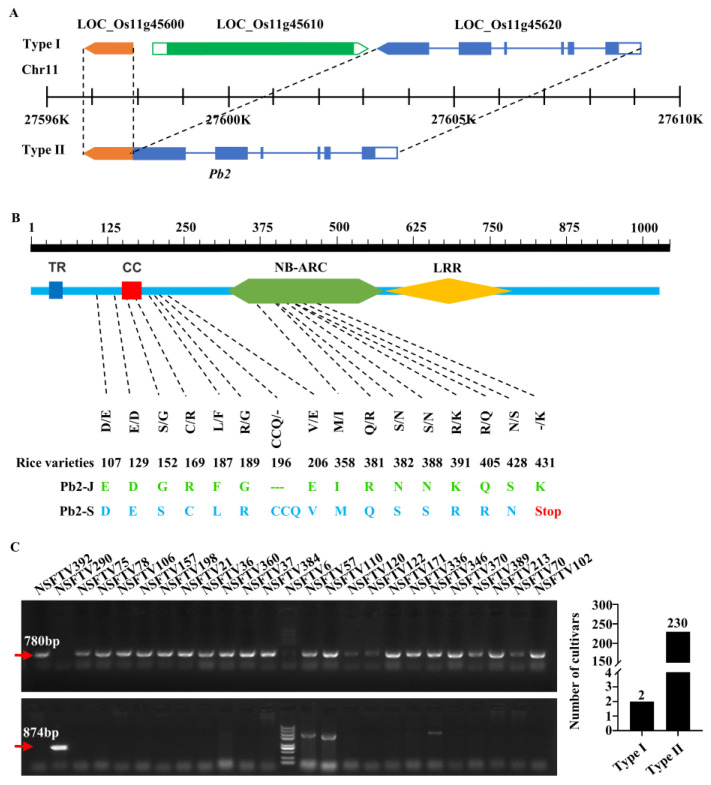
The distribution of *Pb2* in natural populations. (**A**) The arrangement of candidate genes on chromosome 11 in different materials. There are two arrangements: Type I, 3 genes are arranged in sequence, LOC_Os11g45600 is shown in orange, LOC_Os11g45610 (retrotransposon) is shown in cyan and LOC_Os11g45620 is shown in dark blue; Type II, there is no LOC_Os11g45610, and LOC_Os11g45600 is the last exponent of LOC_Os11g45620; the combined gene was named *Pb2*. Arrows indicate gene orientation, blank boxes indicate UTRs and thin lines indicate introns. (**B**) Amino acid sequence and domain analysis of Pb2 in resistant material Jiangnanwan and susceptible material Suyunuo. *Pb2-J*: *Pb2* cloned from Jiangnanwan. *Pb2-S*: *Pb2* cloned from Suyunuo, translation terminated early. TR: transmembrane domain. CC: coiled-coil. NB-ARC: nucleotide binding region of resistance protein. LRR: leucine-rich repeat. (**C**): The arrangement of *Pb2* on chromosome 11 in 232 materials. When only 45600-F/45620-R can amplify the target fragment of 780 bp, it means that there is no LOC_Os11g45610 (top left); when only 45610-F/45620-R can amplify the target fragment of 874 bp, it means that LOC_Os11g45610 is contained (bottom left). On the right are two types of statistics.

**Figure 5 ijms-23-05668-f005:**
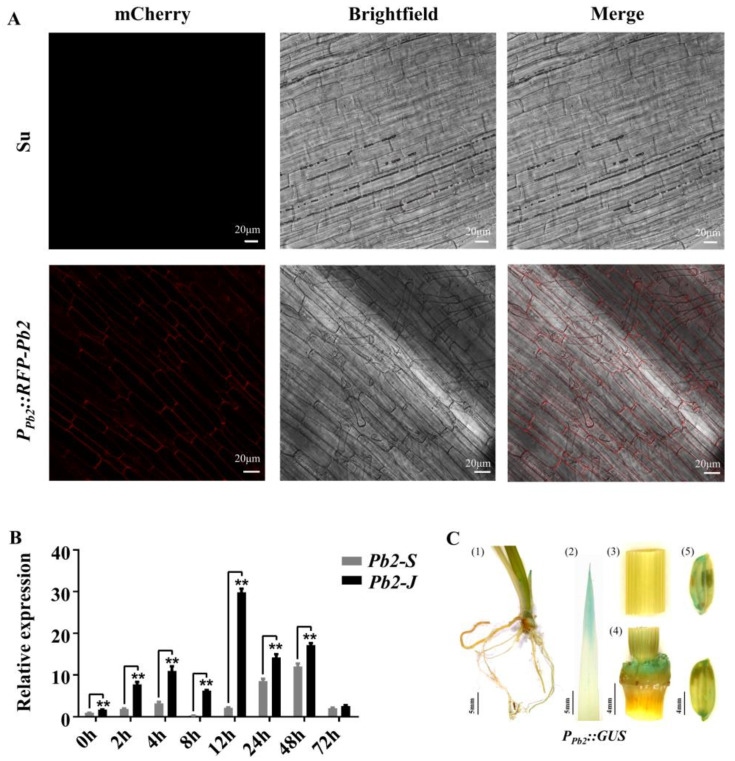
Specific expression of *Pb2*. (**A**) Microscopic observation of *P_Pb2_:: RFP-Pb2* root tip subcellular localization. Pb2 is located to the plasma membrane. The transgenic receptor plant Suyunuo (Su) was used as the control. (**B**) The expression pattern of the *Pb2* gene in landrace Jiangnanwan and Suyunuo inoculated with *M. oryzae Hoku1* was investigated by qPCR. Jiangnanwan and Suyunuo young panicles were collected 0, 2, 4, 8, 12, 24, 48 and 72 h after inoculation. Amplification of *OsActin1* (LOC_Os03g50885) was used as an internal control. The expression level on the ordinate is the relative expression level after excluding the rhythmic interference of sterile water inoculation. Error bars represent standard deviation of three technical replicates; (** *p* < 0.01). (**C**) GUS staining to detect tissue-specific expression. (1) and (2) Staining of roots, leaf sheaths and leaves of three-week-old seedlings of *P_Pb2_::GUS* transgenic material. (3), (4) and (5): Staining of stems, stem nodes and grains of *P_Pb2_::GUS* transgenic materials at heading stage.

**Figure 6 ijms-23-05668-f006:**
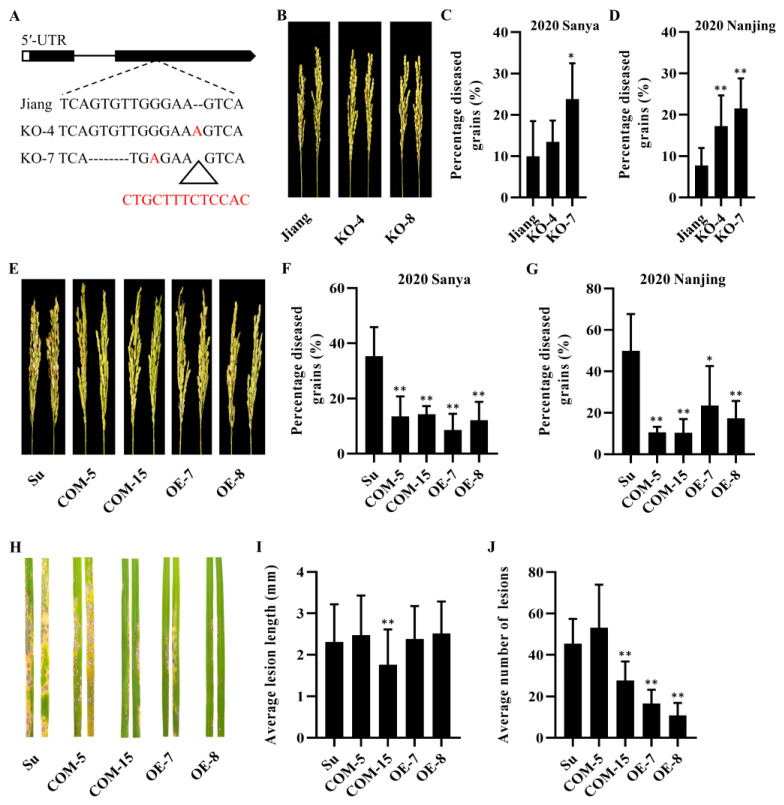
*Pb2* can significantly reduce the number of diseased grains and leaf diseased spots. (**A**) Gene structure of *Pb2* and two types of mutation. (**B**,**E**) Rice panicle three weeks after injection of *Hoku1* spore suspension at booting stage, Nanjing, 2020. Jiang represents the CRISPR-Cas9 transgenic receptor plant and Su represents the complementary and overexpressing transgenic receptor plant. (**C**,**D**,**F**,**G**) Percentage diseased grains (%) of transgenic lines and WT was counted three weeks after inoculation, and each bar represents the average of nine panicles. (**H**) Response of three-week-old seedlings 6 days after inoculation with *Hoku1*. (**I**,**J**) Average lesion length and average number of lesions in WT and transgenic plants. Lesion length and lesion number were measured 6 days after inoculation. Error bars represent standard deviation, and differences between WT and experimental groups were tested with GraphPad Prism 8.3 *t*-test, * *p* < 0.05, ** *p* < 0.01.

**Figure 7 ijms-23-05668-f007:**
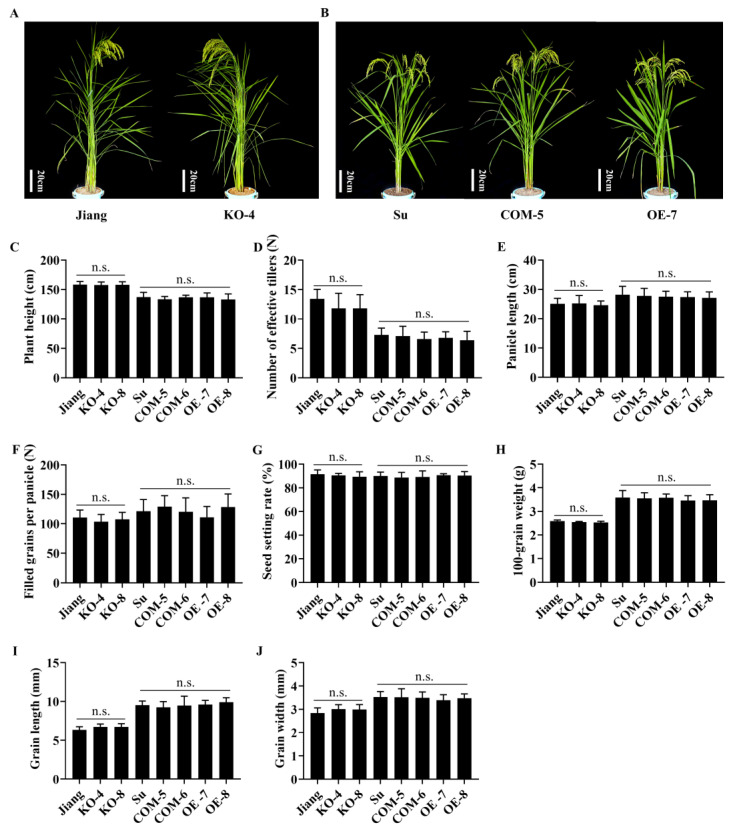
*Pb2* does not affect rice agronomic traits. (**A**,**B**) Field growth status of WT and transgenic plants during filling period. (**C**) Plant height (cm). (**D**) Number of effective tillers (N). (**E**) Rice panicle length (cm). (**F**) Number of grains per panicle. (**G**) Seed setting rate (%). (**H**) 100-grain weight (g). (**I**) Grain length (mm). (**J**) Grain width (mm). Error bars represent standard deviation, differences between WT and experimental groups were tested with GraphPad Prism 8.3 *t*-test. n.s. represents not significant.

**Table 1 ijms-23-05668-t001:** The PBRLs that were associated with panicle blast in 2018 and 2019.

Date	Locus	Chr.	Position	Top SNP ^1^	*p*-Value	Locus Reference ^2^
2018	PBRL-1	1	6852800–7071042	6952618	1.88 × 10^−5^	
2018	PBRL-2 ^3^	3	22459640–22470705	22470705	1.34 × 10^−5^	
2018	PBRL-3	4	32795904–32942350	32937674	1.90 × 10^−5^	*BRF04*, *LABR-45*, *Pikur1*
2018	PBRL-4	6	12756073–13177675	12926253	1.71 × 10^−8^	*Pi-d3*, *Pi-25*
2018	PBRL-5	6	29702212–31175843	29825174	2.24 × 10^−6^	*BRF06-2*, *Pitq1*
2018	PBRL-6	9	11120819–11200238	11120819	3.96 × 10^−6^	*LABR-68*
2018	PBRL-7	11	3414684–3590151	3569620	6.19 × 10^−7^	
2018	PBRL-8	11	7122930–7273845	7173961	1.91 × 10^−6^	
2018	PBRL-9	11	25611850–26487216	25634931	23 × 10^−9^	*SxMBRQTL-1*, *LABR-82*, *LABR-83*, *Pif*, *Pi47*, *Pi-43*
2018	PBRL-10 ^4^	11	27028553–28459253	27235944	8.26 × 10^−10^	*LABR-84*, *LABR-85*, *Pik*, *Pikm*, *Pikp*, *Pi1*
2019	PBRL-11	1	42111551–42745473	42745473	1.66 × 10^−5^	
2019	PBRL-12	3	19109711–20904017	19109711	1.38 × 10^−10^	*LABR-32*
2019	PBRL-13 ^3^	3	22167295–22640834	22679011	4.51 × 10^−6^	
2019	PBRL-14	4	20333322–20595554	20595554	3.10 × 10^−7^	*Pi21*
2019	PBRL-15	6	14977368–16235258	14977368	5.94 × 10^−7^	
2019	PBRL-16	6	22427279–22538034	22463749	1.38 × 10^−6^	
2019	PBRL-17	9	21179219–21643345	21359143	9.84 × 10^−7^	
2019	PBRL-18 ^4^	11	26487216–28465787	26487216	3.49 × 10^−7^	*LABR-84*, *LABR-85*, *Pik*, *Pikm*, *Pikp*, *Pi1*

^1^ Within the detection site, the position of the peak SNP on the chromosome (bp). ^2^ Reported genes or QTLs in the same or overlapping intervals. ^3,4^ Same or largely overlapping genetic regions identified in 2018 and 2019.

## Data Availability

Not applicable.

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
