# Peer review of "Genome-Wide Association Study Identifies a Rice Panicle Blast Resistance Gene, Pb2, Encoding NLR Protein"

_ijms, 2022, doi:10.3390/ijms23105668_

Round 1

Reviewer 1 Report

Dear author

Kindly see attached the comments on your pdf manuscript to improve the quality of your work.

Regards

Author Response

Response to Reviewer 1 Comments

  1. line 31: ...Pb2 “is”..., delete and write “was”

Answer: Thanks for reviewer’s comments. We changed “is” into “was” in P1 line 31 in the revised MS.

  1. line 31: ...expression “is”..., delete and write “was”

Answer: Thanks for reviewer’s comments. We changed “is” into “was” in P1 line 32 in the revised MS.

  1. line 37: ... no “obviously” difference..., delete

Answer: Thanks for reviewer’s comments. We deleted “obviously” in P1 line 37 in the revised MS.

  1. line 37:…It “indicates” that…, replace with indicated

Answer: Thanks for reviewer’s comments. We changed “indicates” into “indicated” in P1 line 38 in the revised MS.

  1. line 70:…“promote each other to”…, delete

Answer: Thanks for reviewer’s comments. We deleted “promote each other to” in P2 line 74 in the revised MS.

  1. line 70:…“together”…, delete and write of the plant to the disease.

Answer: Thanks for reviewer’s comments. We changed “together” into “write of the plant to the disease” in P2 line 74 in the revised MS.

  1. line 111-117: “In this study, we used 230 materials from the Rice Diversity Panel (RDP1) and 2 japonica landraces from the Taihu Lake region to identify panicle blast resistance through injection inoculation method at the booting stage. Based on the GWAS results, a panicle blast resistance candidate gene Pb2 was identified. Through phenotyping of complementation and overexpression transgenic plants, Pb2 was verified to effectively improve the resistance of leaf blast and panicle blast. Our findings provide an important basis for rice blast resistance breeding.” Take this to the Methodology section. Then, state the objectives of the current study

Answer: Thanks for reviewer’s comments and we agree with reviewer’s suggestions. we have removed “In this study, we used 230 materials---for rice blast resistance breeding.” and added “The main goal of this study is to find the panicle blast resistance gene. In previous studies, RDP1 has been widely used in rice blast resistance research, and more than 180 leaf blast resistance-related loci have been identified, of which 16 were identified by natural occurrence in the field [28, 31-34]. However, there is no report on the identification of panicle blast resistance using RDP1. This study provides important clues for the identification and cloning of rice blast panicle genes, and provides an important basis for panic resistance breeding by identifying RDP1 in the field and performing PBRL scanning with GWAS.” in P3 line 118-125 in the revised MS. Otherwise, we also added the source and subpopulation distribution of these 232 accessions in Materials and methods, in P14 line 503-511 in the revised MS and changed “For the panicle blast resistance identification, spore suspension with a concentration of 1×105/mL was used for injection.” into “For the panicle blast resistance identification, spore suspension 2mL with a concentration of 1×105/mL was used for injection at the booting stage of rice.” in P14 line 522-523 in the revised MS.

  1. line 118: Results Before this section, you should write down the 'Material and methods section with all the steps (plant material used, where, when and how the experiment was conducted, data collected, how was GWAS doe, data analysis.

Answer: Thanks for reviewer’s comments. According to reviewer’s suggestion, we added “4.8. Data analysis” and “700-K dataset and efficient mixed-model association (EMMA; TASSEL 5.0) were used for GWAS analysis and significant SNPs were selected -log10P>4 [73]. Haplotype analysis was performed on all SNPs that fell on the gene (including non-significant SNPs). GraphPad Prism 8 software was used for data analysis, t-test at 0.05 level and 0.01 level and least significant difference (LSD) method were used for difference comparison analysis; SPSS 19.0 software was used for correlation analysis.” in Materials and methods in P16 line 604-610 of the revised MS.

  1. line 121-123: “230 varieties were selected from RDP1, including 44 Indica (IND), 55 Temperate Japonica (TEJ), 51 Tropical Japonica (TRJ), 42 AUS, 7 AROMATIC, and other 33 accessions classified as admixtures (ADMIX) between Temperate and Tropical japonica groups (Figure 1B). The injection inoculation method was used to identify panicle blast resistance. ” take this to the methodology section

Answer: Thanks for reviewer’s suggestion. This section has been described in Materials and methods in P14 line 505-509 of the revised MS. So, we decided to delete this part from the “Results”.

  1. line 126:…materials “have” a…, replace with had

Answer: Thanks for reviewer’s comments. We changed “have” into “had” in P3 line 130 in the revised MS.

  1. line 127:…them “are” less…, replace with were

Answer: Thanks for reviewer’s comments. We changed “are” into “were” in P3 line 131 in the revised MS.

  1. line 129:…“In 2018”…, ???

Answer: Thanks for reviewer’s comments. We corrected it and changed “In 2018” into “In the identification results of panicle blast in 2018” in P3 line 133-134 in the revised MS.

  1. line 132-136: “These results indicated that AUS and IND had better resistance to panicle blast, and TEJ had the worst resistance to panicle blast. Since the resistance results of panicle blast were identified in the field and were easily affected by environmental factors, correlation analysis of two year’s data was performed.” Take this to the discussion section

Answer: We agree with reviewer’s suggestions. We have moved “These results indicated that---of two year’s data was performed.” into the “Discussion”, and changed the previous description into “The identification results indicated that AUS and IND had better resistance to panicle blast, and TEJ had the worst resistance to panicle blast. Since the resistance results of panicle blast were identified in the field and were easily affected by environmental factors, but in this study, the results of correlation analysis showed that the identification results of the two years were positively correlated.” in P13 line 421-425 in the revised MS.

  1. line 147-148: “700-K dataset and efficient mixed-model association (EMMA; TASSEL 5.0) were used for GWAS analysis and significant SNPs were selected -log 10 P>4” Take this to data analysis

Answer: We agree with reviewer’s suggestions. We added “4.8. Data analysis” in P16 line 604-610 in the revised MS and moved “700-K dataset---were selected -log 10 P>4” to “Data analysis” in P16 line 605-606 in the revised MS.

  1. line 152-1556: “Seven resistance genes at these four PBRLs have been reported with the details as PBRL-4 (Pid3, Pi25), PBRL-10/PBRL-18 (Pik, Pikm, Pikp, Pi1 ) and PBRL-14 (Pi21) [19, 38, 39]. 6 PBRLs as PBRL-3 (LABR-45), PBRL-6 (LABR-68), PBRL-9 (LABR-82, LABR-83), PBRL-10/PBRL-18 (LABR-84, LABR-85) and PBRL-12 (LABR-32) were co-localized with previous identified QTLs in GWAS analysis of RDP1” take this to the discussion section

Answer: We agree with reviewer’s suggestions and moved “Seven resistance genes---in GWAS analysis of RDP1” into discussion section in P13 line 433-438 in the revised MS and updated references.

  1. line 161:…identified “in” the…, replace with at

Answer: Thanks for reviewer’s comments. We changed “in” into “at” in P5 line 206 in the revised MS.

  1. line 164:…“we selected” 1.47Mb…, delete

Answer: Thanks for reviewer’s comments. We deleted “we selected” in P5 line 208 in the revised MS.

  1. line 164:…and “PBRL18” for…, write 'were selected' after this

Answer: Thanks for reviewer’s comments. We added “we selected” in P5 line 208 in the revised MS.

  1. line 165:…“Combining with the” significant…, delete

Answer: Thanks for reviewer’s comments. We deleted “Combining with the” and changed “significant” into “Significant” in P5 line 209 in the revised MS.

  1. line 166:…“and paired” LD (r 2 ≥0.6)…, replace with paired with

Answer: Thanks for reviewer’s comments. We changed “and paired” into “paired with” in P5 line 209 in the revised MS.

  1. line 190:…“we classified” all SNPs…, delete

Answer: Thanks for reviewer’s comments. We deleted “we classified” and changed “all” into “All”in P6 line 239 in the revised MS.

  1. line 191:…candidate “region” into…, write 'were classified' after this

Answer: Thanks for reviewer’s comments. We inserted “were classified” in P6 line 240 in the revised MS.

  1. line 196:…“we also screened” the candidate…, delete

Answer: Thanks for reviewer’s comments. We deleted “we also screened” and changed “the” into “The”in P6 line 245 in the revised MS.

  1. line 196:…second group“,”… write 'were also screened' before this

Answer: Thanks for reviewer’s comments. We inserted “were also screened” in P6 line 245 in the revised MS.

  1. line 197:…variation “as” Group…, delete

Answer: Thanks for reviewer’s comments. We deleted “as” in P6 line 246 in the revised MS.

  1. line 205-207:…“Haplotype analysis was performed on all SNPs that fell on the gene (including non-significant SNPs), and the results” showed…, take this to the data analysis paragraph

Answer: We agree with reviewer’s suggestions. We added “4.8. Data analysis” and changed “Haplotype analysis was---fell on the gene (including non-significant SNPs), and the results.” into “Haplotype analysis was performed on all SNPs that fell on the gene (including non-significant SNPs).” Incorporated into the increased “4.8. Data analysis” in P16 line 604-605 in the revised MS.

  1. line 210-211:…“we also analyzed all SNPs in the 142.2Kb interval, and 11 genes were identified with” amino…, delete

Answer: Thanks for reviewer’s comments and we agree with reviewer’s suggestions. We deleted “we also analyzed all SNPs in the 142.2Kb interval, and 11 genes were identified with” in P7 line 265-266 in the revised MS.

  1. line 212:…“We performed” haplotype…, delete and write 'The

Answer: Thanks for reviewer’s comments. We changed “We performed” into “The” in P7 line 267 in the revised MS.

  1. line 212:…genes “and found”…, showed

Answer: Thanks for reviewer’s comments. We changed “and found” into “showed” in P7 line 267 in the revised MS.

  1. line 214:…“We further analyzed” the expression…, The analysis of

Answer: Thanks for reviewer’s comments. We changed “We further analyzed” into “The analysis of” in P7 line 269 in the revised MS.

  1. line 218:…“encodes” glycosyl…, replace with encoded

Answer: Thanks for reviewer’s comments. We changed “encodes” into “encoded” in P7 line 273 in the revised MS.

  1. line 218:…which “is”…, replace with was

Answer: Thanks for reviewer’s comments. We changed “is” into “was” in P7 line 273 in the revised MS.

  1. line 219:…It “is” intriguing…, replace with was

Answer: Thanks for reviewer’s comments. We changed “is” into “was” in P7 line 274 in the revised MS.

  1. line 219:…LOC_Os11g45620 “encode”…, replace with encoded for

Answer: Thanks for reviewer’s comments. We changed “encode” into “encoded for” in P7 line 274 in the revised MS.

  1. line 220:…LOC_Os11g45600 “encode”…, replace with encoded for

Answer: Thanks for reviewer’s comments. We changed “encode” into “encoded for” in P7 line 275 in the revised MS.

  1. line 220:…patterns “are”…, replace with were

Answer: Thanks for reviewer’s comments. We changed “are” into “were” in P7 line 275 in the revised MS.

  1. line 221:…“We suspect that” these…, delete

Answer: Thanks for reviewer’s comments. We deleted “We suspect that” and changed “these” into “These” in P7 line 276 in the revised MS.

  1. line 221:…genes “may” be…, delete and write 'were suspected to

Answer: Thanks for reviewer’s comments. We changed “may” into “were suspected to” in P7 line 276 in the revised MS.

  1. line 222:…there “are” three…, replace with were

Answer: Thanks for reviewer’s comments. We changed “are” into “were” in P7 line 277 in the revised MS.

  1. line 223:…Hap.1 “is” associated…, replace with was

Answer: Thanks for reviewer’s comments. We changed “is” into “was” in P7 line 278 in the revised MS.

  1. line 224:…Hap.3 “is” associated…, replace with was

Answer: Thanks for reviewer’s comments. We changed “is” into “was” in P7 line 279 in the revised MS.

  1. line 224:…Hap.2 “is” associated…, replace with was

Answer: Thanks for reviewer’s comments. We changed “is” into “was” in P7 line 279 in the revised MS.

  1. line 241:…“corresponds” to the…, replace with corresponded

Answer: Thanks for reviewer’s comments. We changed “corresponds” into “corresponded” in P9 line 309 in the revised MS.

  1. line 246:…this “is” a common…, replace with was

Answer: Thanks for reviewer’s comments. We changed “is” into “was” in P9 line 314in the revised MS.

  1. line 247:…“we verified” the presence…, delete

Answer: Thanks for reviewer’s comments. We deleted “we verified” in P9 line 315 in the revised MS.

  1. line 247:…natural “populations”…, write 'were verified' before after this

Answer: Thanks for reviewer’s comments. We inserted “were verified” in P9 line 317 in the revised MS.

  1. line 249:…“are” fused together…, replace with showed that

Answer: Thanks for reviewer’s comments. We changed “are” into “showed that” in P9 line 317 in the revised MS.

  1. line 252:…there “are” multiple…, replace with were

Answer: Thanks for reviewer’s comments. We changed “are” into “were” in P9 line 320 in the revised MS.

  1. line 253:…there “is” no…, replace with was

Answer: Thanks for reviewer’s comments. We changed “is” into “was” in P9 line 321 in the revised MS.

  1. line 290:…Pb2 “is” associated…, delete

Answer: Thanks for reviewer’s comments. We deleted “is” in P11 line 365 in the revised MS.

  1. line 291:…“we constructed” complementation…, delete

Answer: Thanks for reviewer’s comments. We deleted “we constructed” in P11 line 366 in the revised MS.

  1. line 293:…cultivar Jiangnanwan“.”…, write 'were constructed' before full stop

Answer: We agree with reviewer’s suggestions. We inserted “were constructed” before full stop in P11 line 368 in the revised MS.

  1. line 295:…Pb2-J “is” relatively…, replace with was

Answer: Thanks for reviewer’s comments. We changed “is” into “was” in P11 line 370 in the revised MS.

  1. line 300:…of the line “have”…, replace with had

Answer: Thanks for reviewer’s comments. We changed “have” into “had” in P11 line 375 in the revised MS.

  1. line 300:…difference “was” detected…, delete

Answer: Thanks for reviewer’s comments. We deleted “was” in P11 line 375 in the revised MS.

  1. line 303:…There “is” no difference…, replace with was

Answer: Thanks for reviewer’s comments. We changed “is” into “was” in P11 line 378 in the revised MS.

  1. line 327:…with “a” greater…, delete

Answer: Thanks for reviewer’s comments. We deleted “a” in P12 line 408 in the revised MS.

  1. line 331:…“has a” high density…, replace with had

Answer: Thanks for reviewer’s comments. We changed “has a” into “had” in P13 line 413 in the revised MS.

  1. line 331:…RDP1 “has a” large…, replace with had

Answer: Thanks for reviewer’s comments. We changed “has a” into “had” in P13 line 413 in the revised MS.

  1. line 333:…“isolates” from…, replace with isolated

Answer: Thanks for reviewer’s comments. We changed “isolates” into “isolated” in P13 line 414 in the revised MS.

  1. line 334:…RDP1 “has a” …, replace with had

Answer: Thanks for reviewer’s comments. We changed “has a” into “had” in P13 line 416 in the revised MS.

  1. line 357:…“We identified multiple” …, replace with Multiple

Answer: Thanks for reviewer’s comments. We changed “We identified multiple” into “Multiple” in P13 line 448 in the revised MS.

  1. line 358:…the PBRL-10 locus“,” …, write 'were identified' before the coma

Answer: Thanks for reviewer’s comments and we agree with reviewer’s suggestions. We inserted “were identified” before the coma in P13 line 448-449 in the revised MS.

  1. line 366-367:…“GWAS is an effective way to identify and clone partial resistance genes [47, 55]” …, Is this important here????

Answer: This is just a transitional sentence, we deleted “GWAS is an effective way to identify and clone partial resistance genes [47, 55]” and updated references in P13 line 457 in the revised MS.

  1. line 400:…“4. Materials and Methods”…, place this section after Introduction

Answer: Thanks for reviewer’s suggestion. According to ijms-template, the Materials and Methods should be placed after Discussion.

  1. line 405:…“we selected” 230 …, delete

Answer: Thanks for reviewer’s comments. We deleted “we selected” and updated references in P14 line 504 in the revised MS.

  1. line 405:…“accessions” from …, write were selected after this

Answer: Thanks for reviewer’s comments. We inserted “we selected” after “accessions” in P14 line 504 in the revised MS.

  1. line 431:…“6” plants for each …, replace with Six

Answer: Thanks for reviewer’s comments. We changed “6” into “Six” in P15 line 532 in the revised MS.

  1. line 439:…script “is” obtained…, replace with was

Answer: Thanks for reviewer’s comments. We changed “is” into “was” in P15 line 540 in the revised MS.

  1. line 440:…the “gwas” file including…, replace with GWAS

Answer: Thanks for reviewer’s comments. “gwas” here means file format. In order to make it easier for understanding and not be confused with GWAS, we deleted “gwas” and changed the sentence “the gwas file including the confidence value (p-value value) of each SNP on the whole genome was generated.” into “get a file containing the confidence value (p-value value) of each SNP on the whole ge-nome.” in P15 line 541-542 in the revised MS.

  1. line 451:…uniformly “dilute” 10…, replace with diluted

Answer: Thanks for reviewer’s comments. We changed “dilute” into “diluted” in P15 line 552 in the revised MS.

  1. line 465:…“promoter” the…, replace with promote

Answer: Thanks for reviewer’s comments. We changed “promoter” into “promote” in P15 line 566 in the revised MS.

  1. line 470:…target “is”…, delete

Answer: Thanks for reviewer’s comments. We deleted “is” in P15 line 571 in the revised MS.

  1. line 483:…that there “is”…, replace with was

Answer: Thanks for reviewer’s comments. We changed “is” into “was” in P16 line 594 in the revised MS.

  1. line 522: “References”, kindly check references and make sure that all cited references are in the text and in correct umber

Answer: Thanks for reviewer’s comments. We have checked all references and made updates.

Reviewer 2 Report

line 55: ....provides resistance to...

line 160: ...with the highest...

line 161: ?? ...at the end...

line 176: ....study of ....???

line 364: is this reference in the list?, ...maybe number of reference to add...

line 386: ...is often accompanied...??

line 416: is this reference in the list?, ...maybe number of reference to add...

line 427: ?? Zhou et al. 2012??? [70]

line 465: ....can promote...

line 466: .were ??? cloned??? and inserted into....OR ...were inserted into...

line 172: italics

Author Response

Response to Reviewer 2 Comments

  1. line 55: ....provides resistance to...

Answer: Thanks for reviewer’s comments. We changed “resistant” into “provides resistance” in P2 line 59 in the revised MS.

  1. line 160: with highest,... replace with “with the highest”...

Answer: Thanks for reviewer’s comments. We changed “with highest” into “with the highest” in P5 line 205 in the revised MS.

  1. line 161: ...identified in the… ?? ...at the end...

Answer: Thanks for reviewer’s comments. We changed “in” into “at” in P5 line 206 in the revised MS.

  1. line 176: ....study of ....???

Answer: Thanks for reviewer’s comments. We changed “study” into “study of” in P5 line 220 in the revised MS.

  1. line 364: is this reference in the list?, ...maybe number of reference to add...

Answer: Thanks for reviewer’s comments. We corrected it and changed “(Mundt, 2014)” into “[54]” in P13 line 457 in the revised MS and we have checked all references and made updates.

  1. line 386: ...is often accompanied...??

Answer: Thanks for reviewer’s comments. We changed “resistance often” into “resistance is often” in P14 line 485 in the revised MS.

  1. line 416: is this reference in the list?, ...maybe number of reference to add...

Answer: Thanks for reviewer’s comments. We corrected it and changed “(Wang et al., 2002)” into “[68]” in P14 line 515 in the revised MS and we have checked all references and made updates.

  1. line 427: ?? Zhou et al. 2012??? [70]

Answer: Thanks for reviewer’s comments. We have checked the original text of “Zhu et al. (2012)” and confirmed that it is “Zhu et al., 2012” and changed into “[70]” in P15 line 528 in the revised MS.

  1. line 465: ....can promote...

Answer: Thanks for reviewer’s comments. We changed “promoter” into “promote” in P15 line 566 in the revised MS.

  1. line 466: .were ??? cloned??? and inserted into....OR ...were inserted into...

Answer: Thanks for your suggestion. We corrected the sentence into “The promoter and the CDS of Pb2 were cloned and inserted into the binary vector pCAMBIA1300s-mCherry to generate PPb2:: RFP-Pb2 for subcellular localization.” In P15 line 567-568 in the revised MS.

  1. line 472: italics

Answer: Thanks for reviewer’s comments. We changed “Agrobacterium tumefaciens” into “Agrobacterium tumefaciens” in P15 line 573 in the revised MS.

Reviewer 3 Report

Dear Authors,

Thank you for your excellent investigation, it was one of the best articles I have reviewed. I have just one recommendartion to decipher the abbreviation NLR in the beginning of the manuscript.

Author Response

Thanks for reviewer’s comments. We added the full name of NLR “Nucleotide-binding domain and Leucine-rich Repeat (NLR)” to the position where NLR in the beginning of the manuscript in P1 line 31 in the revised MS.